# Modeling homogeneous ice nucleation from drop-freezing experiments: Impact of droplet volume dispersion and cooling rates

Ravi Kumar Reddy Addula[*1], Ingrid de Almeida Ribeiro[*2], Valeria Molinero[2], and Baron Peters[1,3]

[1]Chemical and Biomolecular Engineering, University of Illinois at Urbana-Champaign, Urbana, IL 61801, USA.
[2]Department of Chemistry, The University of Utah, Salt Lake City, UT 84112, USA.
[3]Department of Chemistry, University of Illinois at Urbana-Champaign, Urbana, IL 61801, USA.

**Correspondence:** Baron Peters (baronp@illinois.edu)

**Abstract.** Homogeneous nucleation is the prominent mechanism of glaciation in cirrus and other high-altitude clouds. Ice nucleation rates can be studied in laboratory assays that gradually lower the temperature of pure water droplets. These experiments can be performed with different cooling rates, different droplet sizes, and often with a distribution of droplet sizes. We combine nucleation theory, survival probability analysis, and published data on the fraction of frozen droplets as a function of temperature to understand how cooling rate, droplet size, and size dispersity influence the nucleation rates. The framework, implemented in the Python code AINTBAD, provides a temperature dependent nucleation rate on a per volume basis, in terms of approximately temperature-independent prefactor (A) and barrier (B) parameters. We find that less than an order of magnitude dispersion in droplet diameters, if not properly included in the analysis, can cause apparent nucleation barriers to be underestimated by 50%. This result highlights the importance of droplet size-dispersion in efforts to model glaciation in the polydisperse droplets of clouds. We also developed a theoretical framework, implemented in the Python code IPA, to predict the fraction of frozen droplets at each temperature for arbitrary droplet size dispersions and cooling rates. Finally, we present a sensitivity analysis for the effect of temperature uncertainty on the nucleation spectrum. Our framework can improve models for ice nucleation in clouds by explicitly accounting for droplet polydispersity and cooling rates.

## 1 Introduction

The thermodynamics and kinetics of ice formation from water are important for atmospheric science (Koop et al. (2000); Möhler et al. (2007); DeMott et al. (2010); Knopf and Alpert (2023)), preservation of biologically active substances (John Morris et al. (2012); Zachariassen and Kristiansen (2000)), and storage of food products (Goff (1997); Li and Sun (2002)). Nucleation, the first step in ice formation, heralds the onset of important subsequent changes: rapid growth of ice domains (Shultz (2018); Barrett et al. (2019); Sibley et al. (2021)), the release of latent heat (Riechers et al. (2013); Dobbie and Jonas (2001)), and freeze concentration of impurities (Deck et al. (2022); Deville (2017); Stoll et al. (2021)). A quantitative understanding

---

[*]These authors contributed equally to this work

of these processes requires models that accurately predict ice nucleation kinetics. In most applications, the primary source of nuclei is heterogeneous nucleation on various surfaces and impurities at mild supercooling (Alpert and Knopf (2016); Zhang and Maeda (2022); Stan et al. (2009); Kubota (2019)). However, homogeneous nucleation of ice occurs at deep supercooling for pure water droplets in the atmosphere (Koop et al. (2000); Knopf and Alpert (2023); Herbert et al. (2015); Heymsfield and Miloshevich (1993); Spichtinger et al. (2023)) and in laboratory experiments (Murray et al. (2010); Atkinson et al. (2016); Shardt et al. (2022); Riechers et al. (2013); Laksmono et al. (2015)).

Special assays have been developed to study ice nucleation kinetics by monitoring hundreds of small supercooled water droplets (Laval et al. (2009); Shardt et al. (2022); Ando et al. (2018); Tarn et al. (2020)). These experiments provide an independent realization of the nucleation time and/or temperature for each droplet (Tarn et al. (2020); Shardt et al. (2022)). Typically, the kinetics are studied via induction times in isothermal conditions (constant supercooling) (Alpert and Knopf (2016); Herbert et al. (2014); Knopf et al. (2020)) or via the spectrum of ice nucleation temperatures at constant cooling rate (Zhang and Maeda (2022); Ando et al. (2018); Shardt et al. (2022); Murray et al. (2010)). These two types of experiments have important similarities and differences.

For droplets subjected to a constant supercooling, the induction time is exponentially distributed. Several analyses have modeled the exponential decay to understand how nucleation rates depend on supercooling (Alpert and Knopf (2016); Herbert et al. (2014); Knopf et al. (2020)). In experiments where the supercooling is gradually increased, the distribution of nucleation times is more complicated (Murray et al. (2010); Riechers et al. (2013)). Typically, no nucleation events occur until the temperature drops below some critical temperature, and then the nucleation times/temperatures all occur within a focused range (Murray et al. (2010); Riechers et al. (2013); Shardt et al. (2022)). The narrow range of ice nucleation temperatures has motivated the use of a single temperature cutoff for ice nucleation in cloud models (Kärcher and Lohmann (2002)). This approach, however, cannot account for the known impact of cooling rate - which span from about 0.01 to 1 K/min - in the formation of ice in clouds (Stephens (1978); Kärcher and Seifert (2016); Shardt et al. (2022)). Likewise, cloud models typically assume monodisperse distribution of droplet sizes, while the range of sizes of droplets in clouds span typically from 2 to 50 micrometers (Igel and van den Heever (2017)). The combined impact of the cooling rate and droplet size distribution on the analysis of droplet freezing experiments and the prediction of cloud properties has not been, to our knowledge, addressed to date.

To motivate new elements of our approach to introduce cooling rate and droplet polydispersity on the interpretation and prediction of experimental data, we briefly discuss the capabilities and gaps in existing models for analyzing the experiments with steadily cooled droplets. Analyses of drop-freezing experiments can be grouped according to two distinguishing criteria. The first distinction pertains to the models used for interpreting the nucleation rate. Kubota used empirical nucleation rate models (Kubota (2019)), while others have used theoretically motivated rate expressions (often based on classical nucleation theory) (Ickes et al. (2017); Murray et al. (2010); Riechers et al. (2013)). Empirical rate models can provide excellent fits to the nucleation rate data, and successful empiricisms sometimes inspire new theoretical models. However, the fitted rate expressions for nucleation rate from an empirical model lack the interpretability and generalizability afforded by a successful fit to theoretical rate models.

A second distinction pertains to the analysis and interpretation of the droplet nucleation data itself. Some studies focus on the fraction of droplets that nucleate in a specific supercooling range, i.e. the nucleation spectrum (Murray et al. (2010); Shardt et al. (2022); Ando et al. (2018)). The nucleation spectrum has sometimes been interpreted as an intrinsic property of supercooled water and/or the nucleants present in the system (Zhang and Maeda (2022); Alpert and Knopf (2016); Knopf and Alpert (2023)). However, it also depends on variables beyond chemical or interfacial properties, e.g. the cooling rates and droplet diameters. An alternative explanation for the nucleation spectrum begins with the survival probability formalism. In survival probability analyses, the probability that a droplet remains liquid steadily declines with time in proportion to the changing rate of ice nucleation. The survival probability formalism is easily used in combination with theoretical models for the nucleation rate, but the combination remains rare in the ice nucleation literature. Indeed, prior combinations of survival probability and nucleation theory in the ice literature focus on heterogeneous nucleation (Wright and Petters (2013); Marcolli et al. (2007); Alpert and Knopf (2016)).

In this work, we combine survival probability analysis with classical nucleation theory to quantitatively predict the effects of different droplet volumes (Atkinson et al. (2016)) and cooling rates (Shardt et al. (2022)). The experiments that motivated our study observed homogeneous nucleation in narrowly selected droplet diameters, cooled at a steady rate deep into the metastable zone. We are inspired by the experiments to achieve the precise control of droplet diameters, but the atmospheric clouds will naturally have a distribution of droplet diameters (Painemal and Zuidema (2011); Igel and van den Heever (2017)). We demonstrate a method to extract theoretically derived nucleation rate parameters from the experimental survival probability data of monodispersed droplets and droplets with the distribution of diameters. We find that the dispersion of droplet sizes typically found in clouds, if ignored in the data analysis, can cause serious errors in the predicted slopes of nucleation rate vs temperature. These nucleation rate slopes are known to significantly impact the prediction of cloud properties (Herbert et al. (2015); Spichtinger et al. (2023)). To address this, we develop a theoretical framework to predict the fraction of frozen droplets considering an arbitrary dispersion of droplet sizes at any specified cooling rates. We implement this framework in a Python code named IPA, which predicts the fraction of frozen droplets as a function of temperature using as input the distribution of droplet sizes and cooling rates. We expect that this model and its implementation will help improve the accuracy of cloud microphysics predictions by accounting for the natural variability in droplet sizes and cooling rates observed in atmospheric conditions.

## 2 Analytical model to analyze the nucleation of monodispersed droplets

The probability that a single droplet of volume $V$ is not frozen in a given time $t$ can be modelled using the master equation (Cox and Oakes (1984)).

$$\frac{dP(t|V)}{dt} = -P(t|V) \times JV. \tag{1}$$

here $P(t|V)$ is the probability, $J$ is the nucleation rate on a per volume per time basis, and $V$ is the droplet volume. For nucleation rate on a per droplet per time basis, we multiply nucleation rate J by droplet volume(V). Note that J itself is independent

of droplet volume, and accordingly parameters that defines J should also be independent of volume. The temperature is constant and the rate of nucleation in each liquid droplet also remains constant in induction time measurements. On integrating Eq. (1) survival probability becomes $P(t|V) = \exp[-JVt]$. This result has been used to analyze nucleation data in several crystallization studies, e.g. by plotting $\ln P(t|V)$ vs $t$ to estimate $J$ and its supersaturation dependence (Alpert and Knopf (2016); Knopf and Alpert (2023); Stöckel et al. (2005); Kubota (2019); Sear (2014)). In contrast, in experiments where the supercooling increases with time, the nucleation rate in each liquid droplet also increases with time.Peters (2011) The survival probability can be obtained by integrating Eq. (1), we get

$$P(t|V) = \exp\left[-\int_0^t J(t)V\,dt\right],\tag{2}$$

where $P$ is a function of time. However, the data are usually reported as a function of temperature or supercooling (Murray et al. (2010); Shardt et al. (2022)). Since the experiments are conducted at a specific cooling rate $R$ (Murray et al. (2010); Shardt et al. (2022) ), we replace the time variable with temperature using the following relation

$$T = T_m - R \times t,\tag{3}$$

where $T_m$ is the melting temperature. After variable transformation the survival probability becomes

$$P(T|V) = \exp\left[-\frac{V}{R}\int_{T_m}^T J(T')\,dT'\right]\tag{4}$$

Eq. (4) separates protocol-specific factors (droplet diameter and cooling rate) from intrinsic properties of the nucleation kinetics and their dependence on temperature.

Now, we need theoretical models or experimental data for nucleation rate to predict the survival probability. Classical nucleation theory gives the rate for homogeneous nucleation as (Volmer and Weber (1926); Becker and Döring (1935))

$$J = A\exp\left[-\frac{16\pi\gamma^3 v_0^2}{3(\lambda_f^2/T_m^2)k_B T}\left(\frac{1}{T_m - T}\right)^2\right],\tag{5}$$

where $A$ is the kinetic prefactor, $\gamma$ is the interfacial free energy between the ice and water, $\lambda_f$ is the latent heat of freezing, $k_B$ is the Boltzmann constant, $T_m$ is the melting point of ice, $v_0$ is the molar volume of ice and $T$ is the absolute temperature.

The nucleation rate $J$ is a product of the equilibrium concentration of clusters of a critical size and the non-equilibrium flux to post-critical sizes. Classical nucleation theory predicts prefactors and exponential terms with explicit temperature dependencies. The exponent in classical nucleation theory is interpreted as a Gibbs free energy barrier, $\Delta G^*/k_B T$. It depends explicitly on both the absolute temperature and on the supercooling. To account for the temperature dependence of nucleation and the time-dependent temperature, we use $\delta_T = (T_m - T)/T_m$ as the dimensionless temperature and rewrite the expression for $J$ as

$$J = A\exp\left[\frac{-B}{(1-\delta_T)\delta_T^2}\right],\tag{6}$$

here $B = (16\pi\gamma^3 v_0^2)/(3\lambda_f^2 k_B T_m)$ contains shape factors, physical constants, the latent heat, and interfacial free energy. These quantities are nearly independent of temperature for the narrow temperature range where homogeneous nucleation is observed in the experiments (Kashchiev (2000); Sear (2007); Koop et al. (2000)). Thus the parameter $B$ should be a nearly temperature independent parameter, while the barrier $\Delta G^*/k_B T$ is strong function of temperature because of the $1/\Delta T^2$ factor.

The prefactor is related to the frequency at which water molecules at the ice-water interface attach to the critical nucleus and to the number of ice molecules that must attach to surmount the barrier. The prefactor is proportional to the self-diffusivity of water, and therefore it depends on temperature. However, over the small range of nucleation temperatures in this study (ca. 2K) we assume that the prefactor is temperature independent.

Note that both parameters, $A$ and also $B$, are assumed to be temperature independent constants over the narrow range of ice nucleation temperatures. However, they may both differ from values that would be theoretically estimated using properties of ice and water at $T_m$. The value of the interfacial free energy when calculated from the fitted values of $B$ is 33 mN/m at 235.5 K, higher than the interfacial free energy reported by Ickes et al., (i.e. 29.0 mN/m). We show in section 9.2 that by assuming that $A$ is independent of temperature, we transfer its temperature dependence to the effective nucleation barrier.

Using Eq. (6) in Eq. (4), the survival probability becomes

$$P(\delta_T|V) = \exp\left[-\left(\frac{AVT_m}{R}\right)\int_0^{\delta_T} \exp\left(\frac{-B}{(1-\delta_T')\delta_T'^2}\right)d\delta_T'\right]. \tag{7}$$

To our knowledge, Eq. (7) has not been used in previous studies of ice nucleation. It isolates parameter $B$, a property of nucleation kinetics, from the dimensionless group $(AVT_m/R)$. The latter depends on intrinsic properties of ice and water ($A$ and $T_m$) but also on $V$ and $R$ which may be experimental choices or cloud conditions. Eq. (7) is valid for water droplets of volume $V$. In most experiments, there is a distribution of volumes which leads to a distribution of droplet nucleation rates. We consider a distribution of droplet sizes in section 5, but first, we demonstrate that the model can predict the effect of droplet volume for narrowly size-selected droplets.

Across the range of nucleation temperatures observed in experiments (Atkinson et al. (2016); Shardt et al. (2022)) for homogeneous ice nucleation (234 K - 238 K), the factor $(1-\delta_T)$ in the rate expression is always near 0.9. Hence, the nucleation rate expression is approximately $J = A\exp(B'/\delta_T^2)$. Where $B'$ is approximately $B' = B/(1-\delta_T) \approx 1.1B$. With this approximation, we have an analytical solution for the survival probability as follows

$$\ln[P(\delta_T|V)] \approx \left[\frac{A'VT_m}{R}\right]\delta_T \times \left(\frac{\sqrt{\pi B'}}{\delta_T}\mathrm{erfc}\left[\frac{\sqrt{B'}}{\delta_T}\right] - \exp\left[\frac{-B'}{\delta_T^2}\right]\right). \tag{8}$$

To illustrate the use of Eq. (7) and Eq. (8), we analyze one of the survival probability data sets (droplet diameter corresponding to 3.8-6.2 $\mu m$) obtained from Atkinson et al. (2016). Optimized fits of the analytical solution (Eq. (8), with $A' = 1.76 \times 10^{39} cm^{-3}s^{-1}$, $B' = 1.3578$) and the numerical integration (Eq. (7), with $A = 8.68 \times 10^{41} cm^{-3}s^{-1}$, $B = 1.2722$) are shown in Fig. 1. Even though the fits show excellent agreement in both analytical and numerical approaches, we note that a 10 % error in the exponent (from approximating $1 - \delta_T \approx 1.0$ leads to a nearly 1000-fold error in $A'$ and a 10 % error in $B'$ relative

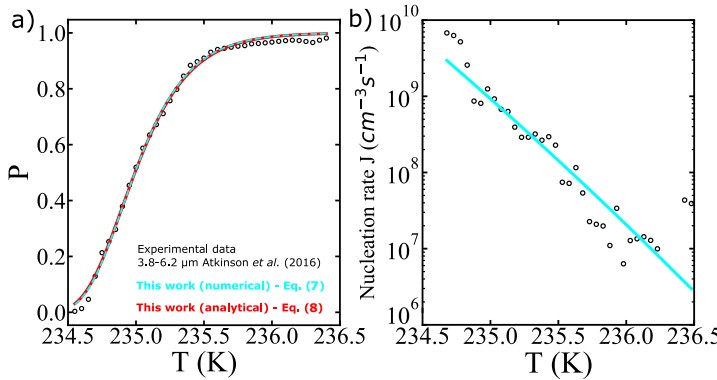

**Figure 1.** a) Illustration of survival probability fits with analytical (red continuous line) and numerical (cyan dashed line) solutions for the integral in Eq. (7). The experimental data represented by the empty black circles used as a reference is from Atkinson et al. (2016). b) Comparison of estimated nucleation rates from experiments using survival probability data as described in Atkinson et al. (2016) (black circles), and the nucleation rate computed using Eq. (7) (cyan line).

to $A$ and $B$). We conclude that precise $A$ and $B$ values require careful treatment of even weak temperature dependencies within $J$. Although the prefactor and barriers are different, the predicted nucleation rates are not. For example, at 234.9 K both approaches give an estimate of nucleation rate to be $1.44 \times 10^9 cm^{-3}s^{-1}$.

The noisy estimates of $J$ in Fig. 1b have been obtained by a finite difference of the cumulative survival probability data. For the finite difference procedure, large numbers of droplets are needed to obtain an estimate of $J$ from the incremental nucleation events in each $\Delta T$ interval. As seen in Fig. 1b, there is considerable noise in the $J$ estimates even in an experiment with hundreds of droplets. Our data analysis approach directly fits a model to the cumulative fraction of frozen droplets. It should therefore remain accurate for data sets with smaller numbers of droplets.

## 155     3    A computer code for analysis of drop-freezing experiments

We implemented the numerical integration in Eq. (7) and analytical model of Eq. (8) in an python code to estimate $A$, $B$ and $J$ from experimental drop-freezing data. The code outputs the parameters $A$ and $B$ from Eq. (7). These are used to compute the nucleation barriers $\Delta G$, the temperature that corresponds to 50% of frozen droplets $T_{50}$, and the homogeneous nucleation rate evaluated at $T_{50}$ using $J_{hom}^{model}(\delta_T) = A exp(-B/[(1 - \delta_T)\delta_T^2])$. The AINTBAD (Analysis of Ice nucleation Temperature

for $B$ and $A$ Determination) code is illustrated in Fig. 2. The code is available in (https://github.com/Molinero-Group/volume-dispersion), last access: 18 Mar 2024).

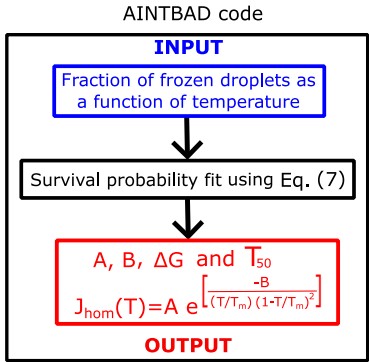

**Figure 2.** Flowchart of the AINTBAD (Analysis of Ice nucleation Temperature for $B$ and $A$ Determination) code.

We use the minimize function from the *scipy.optimize* module in Python to optimize the difference between the target survival probability and the predicted one by adjusting the parameters $A$ and $B$. The chosen optimization method is the Nelder-Mead algorithm, suitable for functions without explicit derivatives. Optional settings include a convergence tolerance of $10^{-4}$ and a maximum iteration limit of 1000.

## 4   Analysis of nucleation spectrum in mono dispersed droplets

Atkinson et al. (2016) monitored the freezing temperatures of narrowly size-selected droplets cooled to temperatures near 235 K at a steady rate of 1.0 $K/min$. A total of 581 droplets were used in the diameter range of 3.8-18.8 $\mu$m, an average of 96 droplets for each diameter. The range of droplet diameters in each experiment and the fraction of droplets that remain at each temperature can be seen as data points in Fig. 3a. We have analyzed the data from Atkinson et al., in two ways. First, we separately fitted the data for each diameter range to Eq. (7). Because the range of diameters each size-selected group is narrow, we have assumed that all droplets in each size range are spheres with the mean diameter for that range. These fits (not shown) result in independent estimates of the optimized nucleation prefactor $A$ and the barrier parameter $B$ from each of the six experiments. Table 1 shows the range of droplet diameters in each experiment, the independent $log_{10}A$ and $B$ estimates, the predicted free energy barrier $\beta\Delta G = B/[(1-\delta_T)\delta_T^2]$ at 235.5$K$, and the predicted nucleation rate (from $J = A\exp[-B/((1-\delta_T)\delta_T^2)]$ at 235.5$K$. The separate $A$ and $B$ estimates vary considerably, but they are highly correlated to each other. Fig. 3b shows $B$ vs. $log_{10}A$ for each of the independent estimates. When $B$ is small (large), $A$ is also small (large). The estimated parameters compensate for errors in each other such that all six data sets yield models that predict consistent nucleation rates. The predicted nucleation rates are shown in Table 1 for the temperature 235.5$K$.

The measurements of Atkinson et al., were all made in the same way, so the same fundamental nucleation rate expression should describe all six size selected data sets. Accordingly, we reanalyzed the data of Atkinson et al., with one global rate expression ($J = A\exp[-B/((1-\delta_T)\delta_T^2)]$), keeping the same $A$ and $B$ values across all six data sets. The nucleation rate parameters obtained from the global fit are $A = 2.79 \times 10^{46}(cm^{-3}s^{-1})$ and $B = 1.45$. Fig. 3a shows the experimental data for different droplet diameters along with model predictions from the global fit. We emphasize that these are six curves, accurately

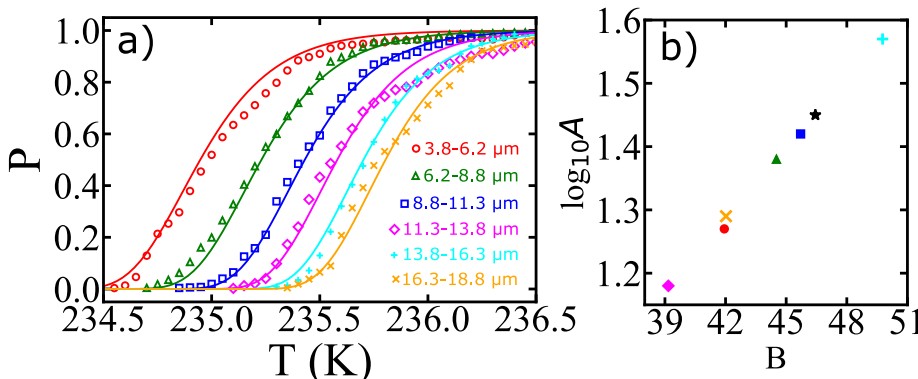

**Figure 3.** a) Illustration of survival probability fits for homogeneous nucleation with varied droplet diameters at a cooling rate of 1 $K/min$. Open symbols represent experimental data from Atkinson et al. (2016) and continuous lines represent model predictions from a global data fit. Colors of symbols indicate different droplet diameters. The individual fits are not shown because the curves overlap closely with the data and globally fitted model predictions. b) Illustration of the correlation between the $\log_{10} A$ and $B$ parameters obtained from individual fits. Black star indicate the ones obtained by the global fit, and other symbols correspond to estimations for different diameters of droplets.

| Droplet diameter ($D/\mu m$) | $\log_{10} A$ | $B$ | $\beta\Delta G_{235.5K}$ | $\log_{10} J_{235.5K}$ |
|:---:|:---:|:---:|:---:|:---:|
| 5.0 | 41.9 | 1.27 | 77.5 | 8.3 |
| 7.5 | 44.5 | 1.38 | 84.2 | 7.9 |
| 10.1 | 45.7 | 1.42 | 86.7 | 8.1 |
| 12.6 | 39.2 | 1.18 | 72.0 | 7.9 |
| 15.1 | 49.8 | 1.57 | 95.8 | 8.1 |
| 17.6 | 42.0 | 1.29 | 78.8 | 7.8 |
| global | 46.4 | 1.45 | 88.5 | 8.0 |

**Table 1.** Computed nucleation rate parameters $A$ and $B$ for various groups of droplet diameters using the volume corresponding to the mean diameter of the group. D is the mean diameter of the droplets in the group in $\mu m$. $\beta\Delta G$ is the free energy barrier for nucleation, and $J$ is nucleation rate computed using fit parameters in Eq. (6). Estimations of $\beta\Delta G$ and $J$ are corresponding to a temperature of 235.5 K.

fitted with just two free parameters, and that both parameters have a clear physical and theoretical interpretation. However, we note that the theoretical relationship between $B$ and $\beta\Delta G$ reflects only the reversible work to create a nucleus at equilibrium, but the parameter $B$ as obtained from experimental data also reflects activation energy contributions from the prefactor. See Section 9.2 for more explanation about this point. Indeed, using the expression $B = (16\pi\gamma^3 v_0^2)/(3\lambda_f^2 k_B T_m)$ and the global fit B = 1.45 in Table 1 results in $\gamma = 33$ mJ/m$^2$ at 235.5 K, which is above the median value of the ice-liquid surface tension in the

literature Ickes et al. (2015).

At a temperature of 235.5 K, the global fit yields a prediction $J = 10^8 cm^{-3}s^{-1}$ for the nucleation rate, consistent with predictions from the independent fits. The free energy barrier at $235.5K$ from the global fit is 88.5 $k_BT$. This is again similar to those obtained from fits to the individual size-selected data sets (Table 1).

Although the rate predictions show remarkable internal consistency, the inferred barriers are scattered and larger than barriers which have been inferred from other data sets (Murray et al. (2010); Shardt et al. (2022); Riechers et al. (2013)). The discrepancy may be a consequence of theoretically unaccounted for temperature dependencies within the pre-factor. Note that the data sets in cyan and pink Fig. 3a actually cross over each other. The crossover indicates that small droplets are nucleating at warmer temperatures than the larger droplets, which should not occur according to nucleation theory. These two anomalous curves correspond to the two most extreme estimates of $A$ and $B$ (upper right and lower left in Fig. 3b). Thus scatter in the $A$ and $B$ parameters, seems to be a true reflection of experiment-to-experiment variation.

Section 5 explores how size dispersity, i.e. the distribution of droplet sizes around the mean diameter, influences the inferred rate parameters. Section 6 examines whether diameter dispersity within the narrow, but non-zero diameter-ranges of Atkinson et al., may still affect the inferred rate parameters.

## 5   Droplets with distribution of volume

Experiments that report on droplet diameter dispersity (Murray et al. (2010); Shardt et al. (2022); Ando et al. (2018)) consistently report a broader range of diameters than the droplets of Atkinson et al. (2016). This section develops a superposition formula to predict the survival probability for experiments with a broad distribution of droplet diameters. We use the term superposition for a data analysis that retains the stratification in freezing temperature, but otherwise pools droplets together regardless of their size. As seen from Fig. 3 and as predicted in Eq. (7) large droplets in a broad distribution will nucleate early (at milder supercoolings), while small droplets will survive to deeper supercoolings. If all temperature dependence comes from the free energy barrier $B/[(1 - \delta_T)\delta_T^2]$, then large droplets that nucleate at milder supercoolings will also nucleate with higher free energy barriers.

The steep sigmoidal survival probabilities for droplets of a specific size, when superimposed, result in a more gradual sigmoid. The gradual sigmoid looks deceptively like the theoretical prediction in Eq. (7), but with artificially reduced barrier $B$ and prefactor $A$ parameters. The analysis here shows how a distribution of droplet diameters broadens the nucleation spectrum, decreasing the inferred nucleation rate barrier.

The joint survival probability distribution with volume and temperature variables is given by

$$P(V, \delta_T) = \rho(V) \times P(\delta_T|V), \tag{9}$$

where $\rho(V)$ is the normalized distribution of droplet volumes and $P(\delta_T|V)$ is the survival probability for droplets of a specific volume, i.e. Eq. (7). The survival probability in time/temperature is obtained by integrating over droplet volumes in the joint

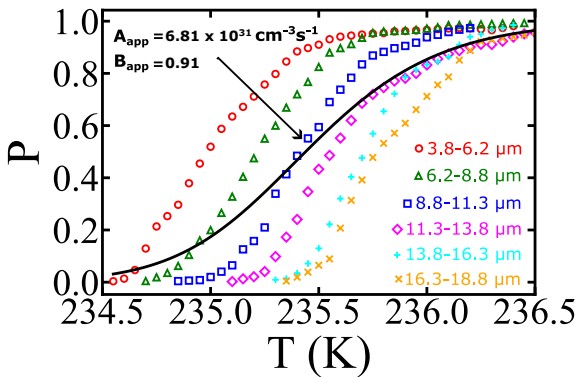

**Figure 4.** Illustrating the effects of the distribution of droplet diameter on survival probabilities. Open symbols are the experimental survival probabilities for different groups of size-selected droplets from Ref. Atkinson et al. (2016). The continuous black line is the superposition survival probability of Eq. (10) from the same kinetics and for a distribution droplets from 3.8-18.8 $\mu m$.

distribution.

$$P(\delta_T) = \int\limits_0^\infty P(V, \delta_T) dV. \tag{10}$$

Here we provide an example calculation to illustrate the effects of a broad droplet volume distribution. Let the normalized (gamma-type) distribution of droplet sizes be $\rho(V) = 8V_0^{-2}V\exp(-2V/V_0)$. Here $V_0$ is the mean volume of the entire range of droplets. Let the survival probability for droplets of any specific diameter be given by $P(\delta_T|V)$ in Eq. (7), with the global fit values of $A$ and $B$, as reported in Table 1. The survival probability for the distribution of droplet volumes can then be obtained using Equations 9 and 10.

We let $V_0 = 1057.1\mu m^3$ in $\rho(V)$ to obtain a distribution with droplets of diameter between 3.0 $\mu m$ and 20 $\mu m$. Note that the model volume distribution spans the range of sizes in the experiments of Atkinson et al. Fig. 4, shows the gradually decreasing survival probability from the superposition as a black solid curve with more steeply changing diameter-selected $P(\delta_T|V)$ data in the background.

If we were unaware of the droplet polydispersity or did not account for it, we might interpret the black curve in Fig. 4, using a survival probability analysis with nucleation theory for droplets correspond to the mean diameter $d$, the volume is computed assuming spherical droplets i.e. $V_0 = (\pi d^3/6)$, thus variations in diameter directly translate to changes in volume. To illustrate how droplet diameter dispersity influences the inferred nucleation rate parameters, we re-optimized $A$ and $B$ now to minimize the residuals between the dispersity superposition result in $P(\delta_T)$ and the naive specific-volume model $P(\delta_T|V_0)$. The resulting $A$ and $B$ values are $6.81 \times 10^{31} cm^{-3}s^{-1}$ and 0.91 respectively. The inferred prefactor ($A_{\text{apparent}}$) is 15 orders of magnitude smaller than that from the global fit of the sets with narrow volume distribution, and the inferred barrier parameter ($B_{\text{apparent}}$) has been reduced by nearly $40\%$. Moreover, the inferred free energy barrier at 235.5 K is now estimated to be $\beta\Delta G = 55.7$ $k_BT$, relative to a value of 88.5 $k_BT$ based on the global fit values to the diameter selected droplet data. The

calculation illustrates how a failure to account for diameter-dispersity causes a spurious broadening of the nucleation spectrum and reduction in the inferred prefactor $A$, barrier parameter $B$, and free energy barriers.

Once we know the variation in droplet diameters, resultant survival probabilities can easily be computed with the help of the python code presented in Section 9 and available in GitHub (https://github.com/Molinero-Group/volume-dispersion). Here variation refers to the overall differences in droplet sizes within a sample, encompassing any deviation from the average droplet diameter. The inputs needed for the program are the proposed distribution of droplet diameters (Gaussian, uniform, gamma, etc.) and the variation of nucleation rate with temperature (see Section 9.1). The output from the code is the effective survival probability.

## 6    How narrow should a droplet distribution be to safely assume a single volume?

First, we ask whether the range of droplet diameters in each experiment by Atkinson et al., each spanning a few $\mu m$, is already broad enough to adversely impact the inferred nucleation parameters. We have considered two test cases for the analysis. One with a midpoint of each reported diameter range as the diameter of all droplets in that group (as shown in the vertical axis of Fig. 5), and the second with a uniform distribution of droplet volumes over the corresponding diameter ranges (as shown in the horizontal axis of Fig. 5). If the range of sizes are sufficiently narrow relative to the mean, then the A and B parameters resulting from a fit to the superposition become identical to those from monodisperse droplets of the mean size. Fig.6 shows that relative width of the volume distribution is sufficient to predict the superposition error. Specifically, for less than 1% error in B ($B_{\text{apparent}}/B_{\text{actual}} > 0.99$) we must have $\Delta V/V < 0.25$. The parity plots for the two values of $A$ and $B$ are presented in Fig. 5. As all the data points are close to the x=y line, we conclude that the droplet diameter ranges in Atkinson et al., are sufficiently narrow to ignore diameter dispersion when inferring the nucleation kinetics.

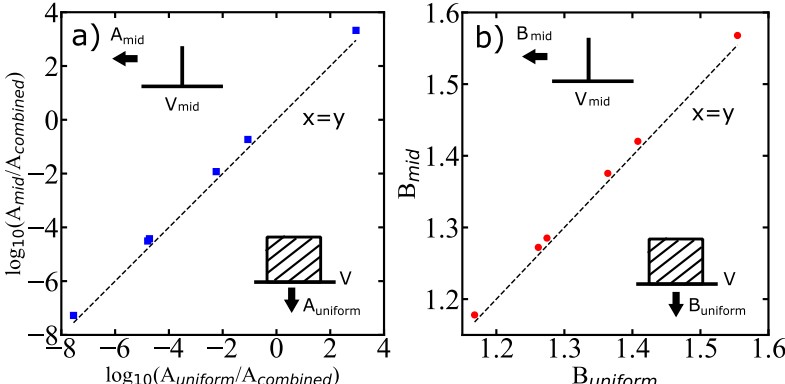

**Figure 5.** Parity plots showing how droplet diameter dispersion influences the inferred $A$ and $B$ parameters. Symbols are inferred $A$ and $B$ parameters with a sharp monodisperse droplet distribution (delta function) corresponding to mean diameter of droplets (mid) as shown in the cartoon on the vertical axis and with a uniform distribution of droplets (uniform) as shown in the cartoon on the horizontal axis. $A_{\text{combined}}$ is the prefactor estimated using global fit. Intervals and midpoints in $V$ correspond to diameter-window of Atkinson et al. The black dashed lines represent $x = y$ lines (parity).

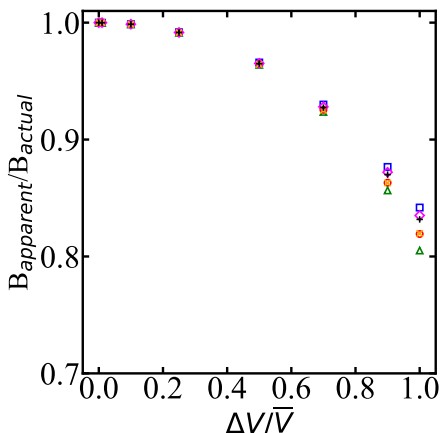

**Figure 6.** When droplet diameter dispersity is ignored, the inferred $B_{\text{apparent}}$ relative to $B_{\text{actual}}$ depends on uniform spread in volume $\Delta V$ relative to the mean volume $\bar{V}$. $\Delta V$ is the width of a uniform distribution of droplet volumes according to the uniform distribution. Symbols of the same color correspond to different widths of uniform distribution and different colors correspond to different mean volume of the droplets. $B_{\text{actual}}$ is the predicted B with monodisperse droplets and $B_{\text{apparent}}$ is the computed $B$ with dispersity in droplet volume.

Fig. 6 shows the ratio between the apparent $B$ parameter from superposition of survival probabilities of droplets with volumes $\bar{V} \pm \Delta V$ and the true $B$ parameter. The analysis shows that the groups with $2\mu m$ variation in diameter resulted in the same nucleation rate parameters with approximately less than $1\%$ variation in estimated free energy barriers. Our analysis in Fig. 6 quantifies the effect of dispersion of droplets in the experiments on predicted $B$ parameters, assuming droplets have uniform distribution. Given $\Delta V/\bar{V}$ and $B$ values from an analysis that imposes volume dispersity, Fig. 6 can be used to estimate

the true value of $B$. The analysis shows that to obtain $B$ within $1\%$ of the correct value, the volume dispersity should be no more than $25\%$ of the mean volume.

## 7   Effect of cooling rate on nucleation parameters

The combined survival probability and nucleation theory expression, as shown in Eq. (7), also predicts that the cooling rate will impact the nucleation spectrum. In this section, we analyze data from Shardt et al. (2022), whose experiments are performed

at two different cooling rates (0.1 $K/min$ and 1.0 $K/min$) with diameter selected droplets of 75 $\mu m$ and 100 $\mu m$. Shardt et al. report the uncertainty in the droplet diameters to be 5 $\mu m$. We model their droplet diameter distribution using a Gaussian with a mean of 75 (or 100) $\mu m$ and a standard deviation of 5 $\mu m$. We have analyzed the survival probability data across the two droplet diameters and two cooling rates with one global fit. The global fit to the survival probability data across the cooling rates and droplet diameters are shown in Fig. 7. The computed nucleation rate parameters from the global fit are

$A = 5.72 \times 10^{28} cm^{-3} s^{-1}$ and $B = 0.81$. The predictions of free energy barriers across the cooling rates and droplet diameters are presented in Table 2.

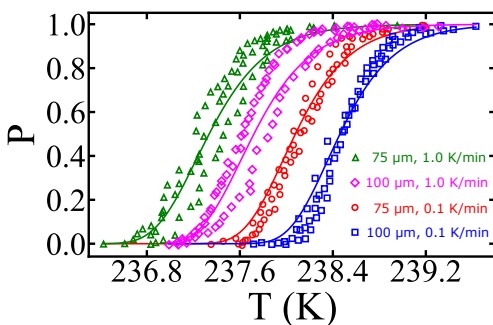

**Figure 7.** Survival probability fits for homogeneous nucleation data with different droplet diameters and cooling rates. Open circles represent experimental data from Shardt et al. (2022) and the solid lines represent model predictions.

| Diameter ($\mu m$) | R ($K/min$) | $\beta\Delta G$ | $T_{50}/K$ |
|---|---|---|---|
| 100 | 0.1 | 57.3 | 238.4 |
| 100 | 1.0 | 55.3 | 237.7 |
| 75 | 0.1 | 56.4 | 238.1 |
| 75 | 1.0 | 54.1 | 237.3 |

**Table 2.** The computed free energy barriers for various droplet diameters across the cooling rates. $T_{50}$ is the temperature corresponding to survival probability of 0.5.

The predictions of $B$ have a similar order of magnitude but are approximately $25\%$ lower when compared to other estimates (Riechers et al. (2013)). We suspect the variation may stem from the difficulties in measuring the precise temperatures of the droplets (Shardt et al. (2022); Tarn et al. (2020); Atkinson et al. (2016)). We also note the computed nucleation rate parameters $A$ and $B$ from Shardt et al., are lower than those from the study of Atkinson et al.. The difference may be due to the uncertainty in droplet temperature measurements, however the two experiments report similar uncertainties in droplet temperatures. Specifically, experiments by Shardt et al. indicated the uncertainty in temperature measurements to be $\pm 0.2K$, and experiments by Atkinson et al. reports $\pm 0.3K$.

## 8 Comparing homogeneous nucleation rate parametrizations

Fig. 8 shows the comparison for the homogeneous nucleation rates using experimental data from Shardt et al. (2022) (blue diamonds) and Atkinson et al. (2016) (green squares). Continuous lines indicate different parametrizations: the fit using the AINTBAD code $J_{hom}^{model}(T)$, where $A = 2.79 \times 10^{46} cm^{-3}s^{-1}$ and $B = 1.45$ for the temperature range of 234.8 to 236.8 K, and $A = 5.72 \times 10^{28} cm^{-3}s^{-1}$ and $B = 0.81$ for 237.0 to 239.1 K (red continuous lines), the parametrization proposed by fitting multiple experimental data $J_{hom}^{equation}(T) = exp[-3.9126T + 939.916]$ Atkinson et al. (2016) (cyan line), as well as the parametrizations based on classical nucleation theory (CNT) from Qiu et al. (2019) (black line) and from Koop and Murray

(2016) (magenta line). In a small temperature range, $J_{hom}^{model}(T)$ captures well the experimental data points. The proposed model, $J_{hom}^{model}(T)$, which works well for micrometer-sized droplets at lower temperatures, may have limitations in accurately capturing the complex nucleation processes occurring in larger droplets at higher temperatures. Thus, $J_{hom}^{model}(T)$ can be used only to predict the homogeneous nucleation rate in the temperature range of the input data used to fit the model.

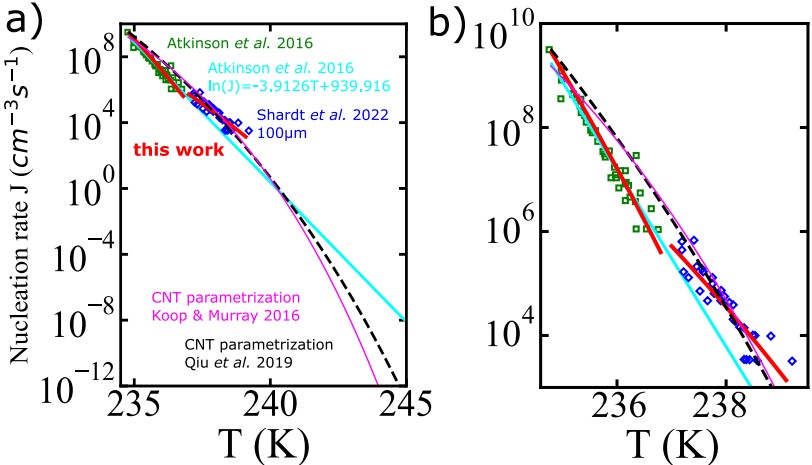

**Figure 8.** a) Comparison of the nucleation rate vs. temperature from experiments of Shardt et al. (2022) (blue diamonds) and Atkinson et al. (2016) (green squares); empirical model proposed by Atkinson et al. (2016) (cyan continuous line); global fit $A$ and $B$ in model $J = A\exp[-B/((1 - \delta_T)\delta_T^2)]$ fitted to Atkinson et al. (2016) and Shardt et al. (2022) (red continuous lines); and the CNT parametrizations from Qiu et al. (2019) (black dashed line) and Koop and Murray (2016) (magenta continuous line). The temperature axis extends to 245 K, the upper plausible limit of homogeneous nucleation temperatures as defined in Herbert et al. (2015). Panel b) shows the nucleation rate in the region from 235 to 240 K only.

All parameterizations predict nucleation rates within an order-of-magnitude of each other and with experiments for temperatures between 235 K and 240 K. However, there is a small gap in the data near 237 K. Fig. 8 suggests a slight disagreement between experimental rates at temperatures below 237 K and those above 237 K. Models for ice nucleation in cloud droplets require nucleation rates that remain accurate over a broad range of temperatures and droplet diameters. Although it is not possible to discriminate between models based on the currently available data, physics-based models should help to build parameterizations that are internally consistent and valid over a broad temperature range.

## 9 A computer code to predict the survival probability using any droplet diameter distribution and cooling rate

We developed a versatile code capable of taking various parametrizations for the homogeneous nucleation rate $J_{hom}(T)$, the droplet diameter distributions (Gaussian, Gamma, uniform, exponential, etc.), and cooling rates to compute the survival probability or fraction of frozen droplets. The code IPA (Inhomogeneous Poisson Analysis) is illustrated in Fig. 9. We use the

nucleation rate data vs temperature as the input to compute the survival probability using the following equation

$$P(\delta_T|V) = \exp\left[-\left(\frac{VT_m}{R}\right)\int\limits_0^{\delta_T} J(\delta_T')d\delta_T'\right]. \tag{11}$$

Eq. (11) is general representation for any given $J$, i.e. it is a version of Eq. (7) that can be used with other nucleation rate models. We evaluate the integral numerically using the trapezoidal rule. Even though Eq. (11) is strictly valid only for a given constant volume of the droplets, we can use Eq. (11) in combination with Eq. (10) to account for the distribution of diameters. Our code includes diverse nucleation rate variations with temperature, including the local parametrization $J_{hom}^{model}(T)$ discussed in the preceding section, the CNT parametrization from sources like Qiu et al. (2019) and Koop and Murray (2016), and the empirical parametrization from Atkinson et al. (2016). Additionally, users can integrate any other parametrization into the code. The code is publicly accessible at (https://github.com/Molinero-Group/volume-dispersion), last access: 18 Mar 2024).

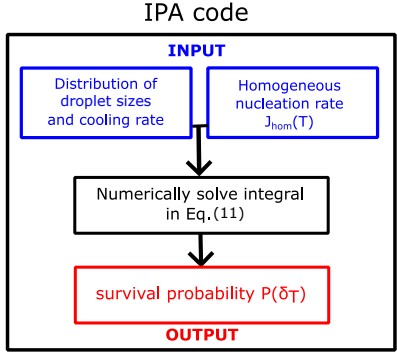

**Figure 9.** Flowchart of the IPA (Inhomogeneous Poisson Analysis) code to obtain the survival probability of water droplets as a function of temperature for any distribution of droplet diameters, cooling rate, and homogeneous nucleation rate parametrization.

### 9.1 Survival probability predictions for cloud data using CNT parametrization

To extend $J_{hom}(T)$ to higher temperatures, and predict freezing for any diameter distribution, we use $J_{hom}^{CNT}(T)$ based on classical nucleation theory (CNT) parametrization of experimental properties of water as previously described in Qiu et al. (2017, 2019). According to CNT, the rate of nucleation is given by

$$J(T) = A(T)\exp\left[\frac{-\Delta G_{hom}}{k_B T}\right], \tag{12}$$

where $T$ is the absolute temperature, $k_B$ is the Boltzmann constant, $A(T)$ is the prefactor, and $\Delta G_{hom}$ is the free energy barrier associated with the formation of a critical ice nucleus. The temperature dependence of the prefactor follows the one of the diffusion coefficient of liquid water using the Vogel–Fulcher–Tammann (VFT) model and was obtained from Koop and Murray (2016). The free energy barrier is formulated as

$$\Delta G_{hom} = \frac{16\pi\gamma_{ice-liq}^3}{3\rho^2\Delta\mu^2}, \tag{13}$$

where $\Delta\mu(T)$ is the excess chemical potential of the liquid with respect to the crystal, $\rho$ is the density of the crystal, and $\gamma_{ice-liq}$ is the surface tension of the ice-liquid interface. We follow the procedure developed by Qiu et al. (2019) to compute the homogeneous nucleation rate $J_{hom}(T)$ as a function of experimental properties of water and ice. In summary, the temperature dependence of the free energy barriers is computed with Eq. (13). The ice-liquid surface tension at the melting point was selected to match the $\gamma_{ice-liq}(T_m) = 31.20\ mJm^{-2}$ to match $J_{hom}$ at $T_{hom} = 238K$ for $\mu L$ droplets cooled at 1 K/min following the experimental data of Atkinson et al. (2016) and Riechers et al. (2013). We approximate the temperature dependence of the ice-liquid surface tension $\gamma_{ice-liq}(T)$ by Turnbull's relation Turnbull (2004) where $\gamma_{ice-liq}(T)/\gamma_{ice-liq}(T_m) = \Delta H_m(T)/\Delta H_m(T_m)$. This parametrization was previously used in Qiu et al. (2019) to study heterogeneous ice nucleation.

Utilizing the CNT parametrization from Qiu et al. (2019) as an input, we integrate it with diverse droplet diameter distributions and cooling rates. The distribution of water droplets in clouds has been examined through the gamma distribution function (Liu et al. (1995); Painemal and Zuidema (2011); Igel and van den Heever (2017)). In Fig. 10a, we showcase how different gamma diameter distributions, manipulated by altering the shape parameter as suggested in Igel and van den Heever (2017), impact droplet diameters. Note that we did not fit a distribution to the data but assume a distribution based on known properties of droplet sizes in clouds. Additionally, Fig. 10b illustrates the survival probability computed via the IPA code using a fixed cooling rate, $q_c$=1 K/min that is typical in clouds Shardt et al. (2022). Notably, the inset reveals a correlation between the shape parameter and freezing temperature. We also include the survival probability for monodisperse droplets, using the most likely diameter from the distributions. Importantly, when the diameter distribution is broader, it has a significant impact on the freezing temperatures.

Typical rates of cooling rates in clouds span from $\sim$ 0.01 to 1 K/min (Stephens (1978); Kärcher and Seifert (2016); Shardt et al. (2022)). Furthermore, Fig. 10c shows that by each tenfold increase in the cooling rate decreases $T_{50}$ by approximately 0.5 K, supporting that an explicitly account of the cooling rate is important in the modeling of ice nucleation. These analyses indicate that explicit account of the cooling rate and droplet size distribution are important for accurate modeling of cloud micro-physical properties.

## 9.2 $B$ as obtained from experiment reflects both diffusion and nucleation barriers

Each of the two data sets, Fig. 3 from Atkinson et al.. and Fig. 7 from Shardt et al., follow the theoretically predicted trends in cooling rate and droplet size dependence. When compared to each other, the larger droplets of Shardt et al., as expected, nucleate at higher temperatures than those of Atkinson et al. Therefore, there is no discrepancy between theoretical expectations and the directly monitored nucleation temperatures. However, the estimated Gibbs free energy barriers from the data of Shardt et al. are smaller than those estimated from the data of Atkinson et al. If $\Delta G^*/k_B T = B/(T * \Delta T^2)$ with constant $B$, then $\Delta G^*$ should be larger for the droplets of Shardt et al. which nucleate at higher temperatures. We have shown that size dispersion causes an underestimation of the B and A parameters. As detailed in section 10, noise in the temperature measurements can also broaden the distribution of nucleation temperatures, causing a similar underestimation of B, A, and $\Delta G^*$. However,

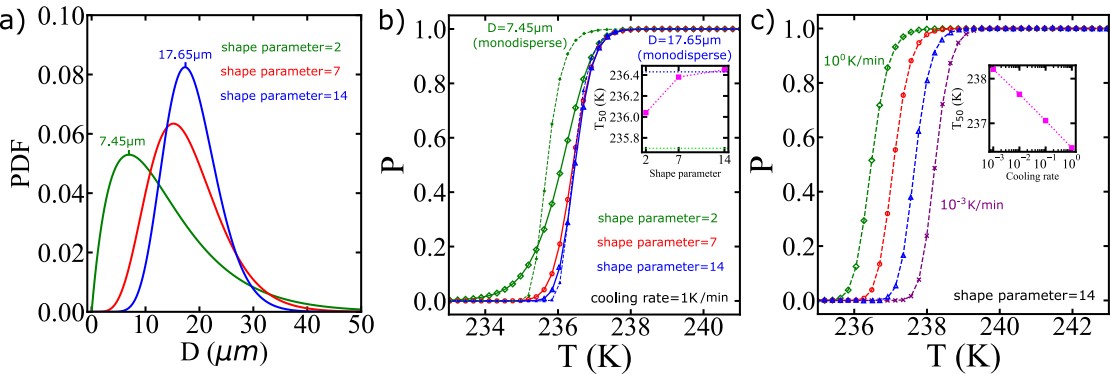

**Figure 10.** We use the CNT parametrization from Qiu et al. (2019) as an input, and we integrate it with diverse droplet diameter distributions and cooling rates. a) We employ a gamma distribution characterized by different shape parameters as used in Igel and van den Heever, where the most likely diameter is indicated for shape parameters 2 and 14. b) The survival probability predictions for the distributions shown in a) and cooling rates $q_c = 1$ K/min. The dashed lines represent the survival probability considering the case of droplets with a single diameter with the same values as the most likely values in a). The inset shows the temperature corresponding to a survival probability of 0.5 using the droplet diameter distribution with different shape parameters, with horizontal dotted lines indicating the results using monodisperse droplets. c) The survival probability predictions using the same droplet diameter distribution but with varying cooling rate.

uncertainty in droplet temperatures cannot explain why the $\Delta G^*$ obtained from Shardt et al. are smaller, as that data set has lower uncertainty ($\pm 0.2K$) than the data set from Atkinson et al. ($\pm 0.3K$).

Alternatively, larger nucleation barriers at the lower temperatures (larger supercoolings) may result from combined effects
of the nucleation barrier (estimated with the AINTBAD code) and diffusion barriers within the prefactor (not yet considered). We analyze this possibility by first predicting the survival probability of liquid droplets upon cooling for a proposed narrow distribution of droplet volumes using the IPA code with a cooling rate of 1 K/min and the $J_{hom}^{CNT}(T)$ CNT parametrization from Qiu et al. (2019) (Fig. 11b and Fig. 11c) and then analyzing these synthetic survival probabilities with the IPA code to extract the effective barrier from the B parameter. Fig. 11a shows that the values of $\Delta G$ at $T_{50}$ obtained from the AINTBAD code
align closely with the sum of free energy barriers for diffusion and homogeneous nucleation of the CNT parametrization of Qiu et al. (2019). This suggests that the high effective barriers for the smaller droplets may originate on a steeply increasing barrier for diffusion. Such an increase is not represented in the parametrization of Qiu et al. (2019) or in Koop (Koop and Murray (2016)), which model the temperature dependence of the diffusion using the Volger-Fulcher-Tamman (VFT) equation with $T_0$ = 148 K, while recent experiments support a steeper decrease in the self-diffusion coefficient $D(T)$ of water as it approaches
its maximum in isobaric heat capacity at 229 K (Pathak et al. (2021)). We interpret that the increase steepness of $D(T)$ on approaching the temperature of maximum heat capacity could be responsible for the larger apparent barrier obtained from the AINTBAD fits to the experimental data. Our finding support a reassessment of the temperature dependence of the prefactor in the parameterizations of the homogeneous ice nucleation rates based on the most current experimental data.

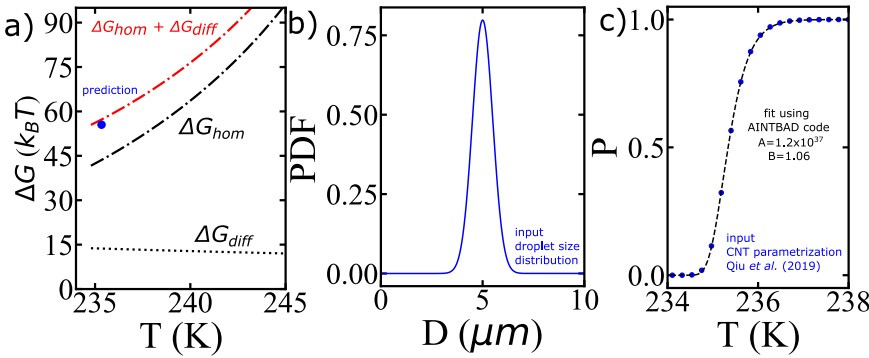

**Figure 11.** a) Comparison between the nucleation barriers computed using the CNT parametrization from Qiu et al. (2019) and the predicted one from the model. b) Distribution of droplet diameters, c) survival probability prediction using the IPA code and the CNT parametrization from Qiu et al. (2019) (blue circles) and the fit using the AINTBAD code (black dashed line). The narrow distribution essentially behaves as a delta distribution.

## 10 Impact of temperature uncertainty on the apparent nucleation barriers

Another important factor that has a significant effect on the measured nucleation spectrum is the measurement of droplet temperature. The estimations of the droplet temperatures in freezing experiments show large variability (Tarn et al. (2020); Shardt et al. (2022)). The highest level of accuracy in the temperature measurements is $\pm 0.2K$ (Shardt et al. (2022)).

In this section, we conduct a sensitivity analysis using our model to quantify the impact of temperature measurement uncertainty on estimated free energy barriers. To perform this analysis, we utilized data from frozen 75 $\mu$m droplets, as

presented in Shardt et al. (2022), which were collected at a cooling rate of 0.1 K/min. Through the HUB-backward code from de Almeida Ribeiro et al. (2023), we determined the optimized differential spectra denoted as $n_m(T)$ based on the frozen fraction (represented by the red continuous line in Fig. 12a). The resulting parameters derived from this analysis were $T_{\text{mode}} = 238.2$K and $s = 0.33$, where $T_{mode}$ represents the most probable freezing temperature within the distribution, and $s$ characterizes the distribution's spread. Subsequently, we employed the original distribution (red continuous line in Fig. 12b)

to generate random temperature values, augmenting them with random values drawn from a uniform distribution within the range of -0.4 to +0.4 K (or -0.2 to 0.2 K). These additional values introduce noise into the data. We sampled a total of 100 temperature values, equivalent to simulating the behavior of 100 droplets in an experimental setup.

The resulting differential freezing spectra are illustrated by the blue squares and green triangles in Fig. 12b. For each case, we calculated the survival probability and fitted the data using Eq. (7), resulting in the continuous lines depicted in Fig. 12c.

390 The effects of temperature variation on the nucleation spectrum are summarized in Table 3. We conclude that measurements with $\pm 0.2K$ and $\pm 0.4K$ variations, resulted in $8\%$ and $14\%$ variation in the computed free energy barriers, respectively. Even though the predictions of free energy barriers show a strong dependence on the uncertainty in temperature measurements, the

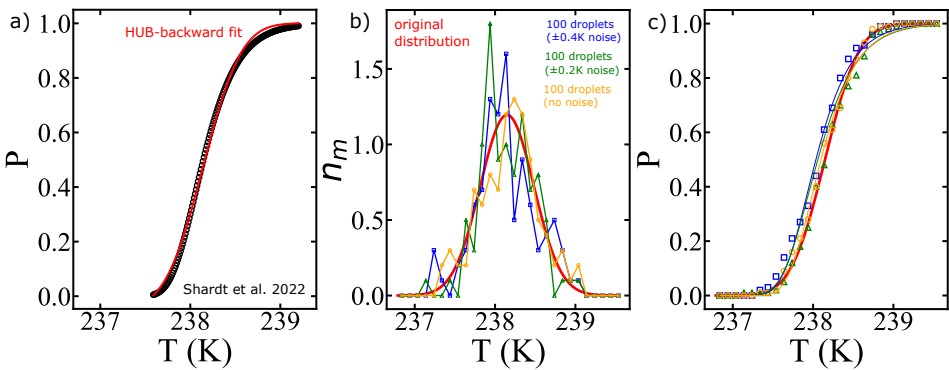

**Figure 12.** Illustration of the effect of temperature uncertainty on the nucleation spectrum. a) We use the HUB-backward code, (de Almeida Ribeiro et al. (2023)) to find the best fit (continuous red line) of the experimental data Shardt et al. (2022) (black circles). b) The continuous red line shows the optimized differential spectra denoted as $n_m(T)$, with resulting parameters $T_{\mathrm{mode}} = 238.2K$ and $s = 0.33$, where $T_{\mathrm{mode}}$ represents the most probable freezing temperature within the distribution, and $s$ characterizes the distribution's spread. We use the red continuous line as the original distribution to generate 100 temperature values (orange circles). Random noise is introduced in the temperature ($\pm 0.4$ K shown in blue squares, $\pm 0.2$ K shown in green triangles). c) The survival probability fit (continuous lines) of the artificially generated data based on Eq. (8).

nucleation rates are insensitive as shown in Table 3. However, the noise in the predicted rate data increases with the noise in temperature measurements.

| Noise $(K)$ | B | $\beta\Delta G$ | $\log_{10}(J_{238.2K})$ |
|---|---|---|---|
| 0.0 | 0.97 $\pm$0.05 | 67.0 $\pm$3.0 | 5.06 $\pm$ 0.03 |
| 0.2 | 0.89 $\pm$0.06 | 62.0 $\pm$4.0 | 5.01 $\pm$ 0.04 |
| 0.4 | 0.83 $\pm$ 0.03 | 58.0 $\pm$2.0 | 5.05 $\pm$ 0.07 |

**Table 3.** The computed free energy barriers and nucleation rates for various uncertainties in the temperature measurements. Mean and standard deviation of B and $\beta\Delta G$ are computed from 5 different estimates. We consider $T_{50}$ to be 238.2K for all the cases presented in this table.

## 11  Conclusions

Homogeneous ice nucleation is the predominant mechanism of glaciation in cirrus and other high-altitude clouds, making the accurate representation of cloud microphysics highly dependent on the homogeneous nucleation rates. It has been shown that the predictions of cloud models are sensitive to the rate of cooling and variations in the slope of the nucleation rate with temperature (Herbert et al. (2015)). In this study, we demonstrate that cooling rate and dispersion in droplet diameters can

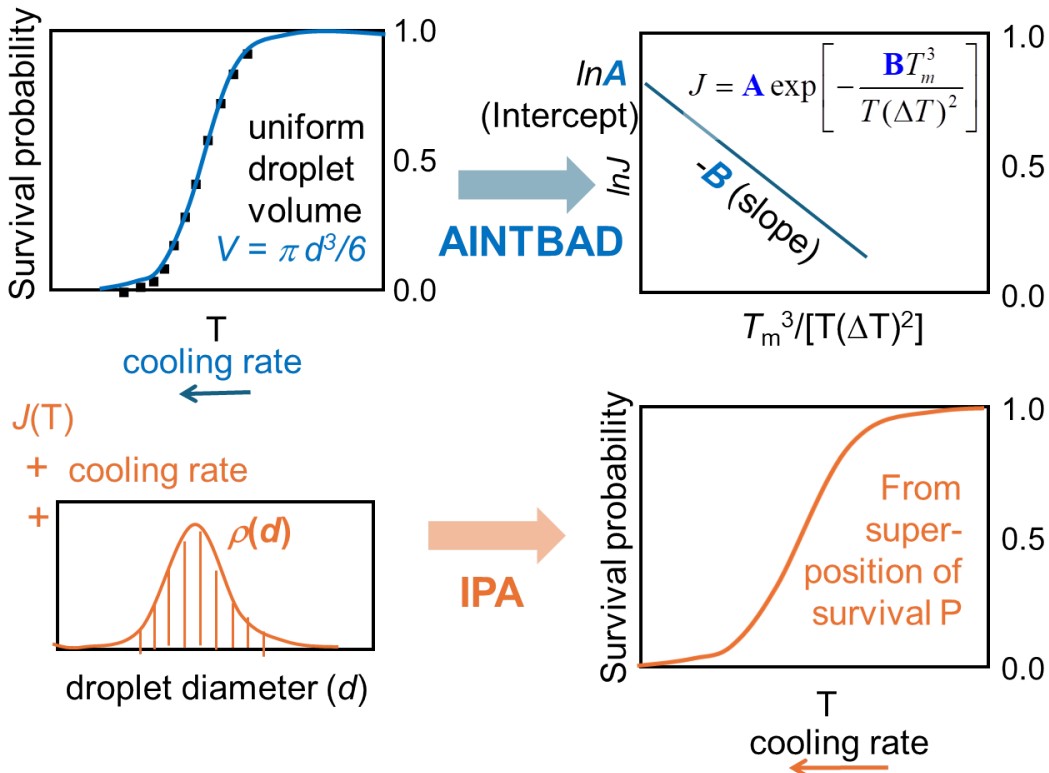

**Figure 13.** Diagram illustrating the usage of the AINTBAD and IPA codes.

lead to substantial changes in the freezing temperature of droplets, stressing the need to incorporate these variations into cloud models.

Homogeneous nucleation rates can be obtained from experiments that record the freezing temperature as pure water droplets are gradually cooled to temperatures far below $0^0C$. Prior studies analyzed these experiments using Poisson statistics to infer rates at different temperatures, and then fitted the rate vs temperature data to empirical or theoretical rate expressions. Here we directly analyze the fraction of frozen droplets vs temperature data to estimate rate parameters within a stochastic survival probability framework. We implement our approach in a Python code, AINTBAD (Fig. 13) that extracts the prefactor $A$ and barrier $B$ parameters according to a classical nucleation theory (CNT)-type rate expression. An advantage of our method is that it does not require large number of droplets to estimate accurate nucleation rates at each temperature. Although AINTBAD does not directly use nucleation rates in the optimization, it yields accurate estimates of the nucleation rate and avoids the noise associated with numerical differentiation.

We applied the AINTBAD code to analyze the homogeneous nucleation data obtained from two different studies: Atkinson et al. (2016); Shardt et al. (2022). We first used the new framework to extract parameters and rate expressions from experiments

on six groups of diameter selected droplets, from $5.0 \pm 1.2 \mu m$ to $17.5 \pm 1.2 \mu m$. The analysis gave similar prefactors, barriers and rates across all six experiments. We further showed that all six experiments can be fitted with just two parameters from one global parametrization. The results of Shardt et al. including four experiments with two droplet sizes and two cooling rates were similarly fitted with a single pair of $A$ and $B$ values.

We derived a superposition formula to predict how a distribution of droplet sizes causes a broadening in the distribution of nucleation temperatures. The broadening causes the AINTBAD analysis to underestimate the $B$ parameter relative to that obtained from monodisperse droplets of the same mean size. For example, pooling droplets with a broad range of diameters from 4 to 19 micrometers significantly reduces the slope in the fraction of frozen droplets with temperature and leads to a ~40% reduction of the inferred barrier. The erroneous parameterization would result in large errors for glaciation rates in cloud microphysics models. Accurate parameterizations can be obtained from experiments like those of Atkinson et al. (2016); Shardt et al. (2022), in which the droplet size distributions are sufficiently narrow to yield parameters that are indistinguishable from perfectly monodisperse droplets.

Our analysis suggest that the barrier obtained with AINTBAD is the sum of the nucleation and diffusion barriers. As the nucleation barriers decrease monotonously with temperature, we infer that the increase in overall barrier for nucleation at 235.5 K originates in a steeper temperature dependence of the diffusion coefficient of water $D$ that controls the temperature dependence of the prefactor. Our interpretation is consistent with the steep decrease in $D(T)$ unveiled by experiments on water approaching the maximum in isobaric heat capacity at 229 K (Pathak et al. (2021)), and calls for a reassessment of the representation of $D(T)$ in CNT parametrizations of ice nucleation rates.

While laboratory experiments strive to study nucleation in the narrowest possible distribution of droplet sizes to avoid spurious impacts on the parametrization of nucleation rates, clouds can have a relatively broad distribution of size of water droplets. We develop the Python code IPA to predict the nucleation spectrum for any given distribution of droplet diameters at any cooling rate. IPA uses as input the distribution of droplet diameters and a parametrization of nucleation rate $J(T)$ from the literature (Fig. 13). IPA includes various previously reported parameterizations and can be extended to use others introduced by the users. We have demonstrated the application of IPA in predicting the impact of droplet diameter distributions typical of clouds on the evolution of the fraction of frozen droplets with temperature. By integrating the cooling rate and size dependence into the ice nucleation rates, the results and tools provided in this study could be used to improve and test approximations made in cloud models.

We restrict our discussion in this article to homogeneous nucleation, but it might be possible to develop similar methods for analysis of heterogeneous nucleation data. A key challenge is that pure water droplets vary only in volume, while heterogeneous nucleation sites may vary in surface chemistry, pore geometry, and size (area) of the active region. These differences lead to sites with different barriers and also different prefactors. Except for special cases of highly regular surfaces, the estimated $A$ and $B$ parameters will then reflect a superposition of survival probabilities from many different types of sites. To illustrate this point, we analyze the data of fraction of ice vs temperature for ice nucleation by Kaolinite from Zhang and Maeda (2022) using the AINTBAD code. The estimated barrier at $T_{50} = 267.2$ K is approximately $2\,k_B T$ (Supp. Fig. S1), a low value indicative of a

superposition with nucleation sites of many different barriers. Further developments are needed to disentangle the contributions of different sites in heterogeneous nucleation of ice.

*Code availability.* The codes and data that support the findings of this study are available in the github repository https://github.com/Molinero-Group/volume-dispersion

*Author contributions.* RKRA, IdAR, VM, and BP designed the project and prepared the manuscript. RKRA, IdAR, and BP developed the models and performed the analysis.

*Competing interests.* The contact author has declared that none of the authors has any competing interests

*Acknowledgements.* We thank Geoffrey Poon, Max Flattery, and Conrad Morris for helpful discussions. This work was supported by the Air Force Office of Scientific Research through MURI Award FA9550-20-1-0351.

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
