# Peer review of "Modeling homogeneous ice nucleation from drop-freezing experiments: Impact of droplet volume dispersion and cooling rates"

_EGUsphere, 2024_

## Author Comment (AC1)

We thank both reviewers for their thorough reading of the manuscript and thoughtful suggestions. Below we address each of the reviewer's questions and suggestions.

Reviewer comments are in black.

Responses are in blue.

Changes to the manuscript are shown in red here and in the annotated pdf for review.

**Reviewer #1**

The manuscript submitted by Addula et al., describes the development of a model to study homogeneous ice nucleation (nucleation rates, survival probabilities) for micrometer sized droplets. The model is based on the classical nucleation theory (CNT) and survival probability and optimizes two parameters (preexponential factor and energy barrier) for the representation of homogeneous ice nucleation based on experimental data obtained from Atkinson et al., 2016 and Shardt et al., 2022. The presented python codes (AINTBAD and IPA) seem to work properly on optimizing model parameters and predicting freezing behaviors. Furthermore, the authors expand their model to illustrate how droplet size variations and different cooling rates influence homogeneous ice nucleation. The presented work has implication in atmospheric chemistry and physics by explaining homogeneous ice nucleation in droplet experiments and clouds. The authors shared their code on github, which is very valuable for the community. The writing is quite concise and precise; however, some important information is missing, especially to guide unfamiliar readers through the manuscript. I believe that the model has a high potential to help fundamentally understand homogeneous ice nucleation and could even be developed further to analyze heterogeneous ice nucleation data. Therefore, I recommend this paper for publication in ACP after the authors address the following comments and questions.

Major comments

Link your results to literature and physical meaning: As the authors state in line 97 the parameters A and B are temperature-independent parameters and B/[(1-deltaT)/deltaT^2] represents the free energy barrier of nucleation. What physical meaning does the pre-exponential factor represent? What values would you find for the properties of ice and water (e.g. interfacial free energy) when back calculating from the fitted values for B (e.g. from the global fit from Table 1)? Would these values line up with literature, e.g. Ickes et al., 2015?

We clarified the physical meaning of our model parameters and their temperature dependence below eq. 6 with this addition:

$B=16\pi\gamma 3v_0^2/3\lambda_f^2 k_B T_m$ contains shape factors, physical constants, the latent heat, and interfacial free energy. These quantities are nearly independent of temperature for the narrow temperature range where homogeneous nucleation is observed in the experiments (Kashchiev (2000); Sear (2007); Koop et al. (2000)). Thus the parameter B should be nearly temperature independent, while the barrier $\Delta G^*/k_B T$ is a strong function of temperature because of the $1/\Delta T^2$ factor.

The prefactor is related to the frequency at which water molecules at the ice-water interface attach to the critical nucleus and to the number of ice molecules that must attach to surmount the barrier. The prefactor is proportional to the self-diffusivity of water, and therefore it depends

weakly on temperature.  However, over the small range of nucleation temperatures in this study (ca. 2K) we assume that the prefactor is temperature independent.

Note that both parameters, A and also B, are assumed to be temperature independent constants over the narrow range of ice nucleation temperatures.  However, they may both differ from values that would be theoretically estimated using properties of ice and water at $T_m$.  The value of the interfacial free energy when calculated from the fitted values of B is 33 mN/m, higher than the interfacial free energy reported by Ickes et al (2015)., (29.0 mN/m).  We show in section 9.2 that by assuming that A is independent of temperature, we transfer its temperature dependence to the effective nucleation barrier.

Regarding the value of physical properties that would be obtained from the values of B in the global fit of Table 1, we now explain in the text below Table 1 that a literal interpretation of B assuming the expression from CNT and neglecting the fact that it encloses also the temperature dependence of the prefactor renders a value of the ice-liquid surface tension that is  high compared to the bulk of values proposed in the literature (for which we cite the excellent compilation of Ickes).  See pg. 8 below Table 1 for additions:

Indeed, using the expression $B=16\pi\gamma 3v_0^2/3\lambda_f^2 k_B T_m$ and the global fit B = 1.45 in Table 1 results in $\gamma$=33.0 mJ/m$^2$ at 235.5 K, which is above the median of the values accepted for the ice-liquid surface tension in the literature by Ickes et al. (2015).

Define the variation in droplet sizes and size distributions better: I got confused with the droplet size variations since the authors use the terms variation, dispersity, and delta V interchangeable through the manuscript (e.g. section 5 and 6). What does delta V, dispersity and variation mean? How does size distribution (diameter?) link to volume distributions (function of r$^3$)? A clear definition of the variation parameter would be very helpful, maybe even plotting a size distribution from one of the analyzed datasets and showing how the fitted distribution matches the data (if possible).

Thank you for urging us to clarify the meanings of $\Delta$V, dispersity, and variation. We always refer to spherical droplets with diameter $d$. $\Delta$V represents the spread in the volume distribution. We added the following lines in section 5:

The volume is computed assuming spherical droplets i.e. $V_0=\pi d^3/6$, thus variations in diameter directly translate to changes in volume.

We also added definitions after each term.  On page 10:

Here variation refers to the overall differences in droplet sizes within a sample, encompassing any deviation from the average droplet diameter.

And on page 9:

… size dispersity, i.e. the distribution of droplet sizes around the mean diameter, …

Regarding the droplet size distribution in clouds we clarified with this text on pg. 16:

*Note that we did not fit a distribution to the data but assume a distribution based on known properties of droplet sizes in clouds.*

Expand the implications: I wish the authors could expand their implications in the abstract, section 9.1, and conclusions. To me it was not clear what we can learn from the model about homogeneous ice nucleation in the atmosphere. For example, the authors show in Figure 10c that the cooling rate influences the T50 values, however how does that relate to cooling rates observed in the atmosphere at different uplifts? I think the paper would be more suitable for ACP when the authors link their results better to implications in atmospheric science.

We have revised the manuscript to expand the implications of our findings in the abstract, section 9.1, and conclusions, particularly focusing on their relevance to atmospheric science. We rewrote the whole abstract:

[revised manuscript text omitted]

We also added to the conclusion

While laboratory experiments strive to study nucleation in the narrowest possible distribution of droplet sizes to avoid spurious impacts on the parametrization of nucleation rates, clouds can have a relatively broad distribution of size of water droplets. We develop the Python code IPA to predict the nucleation spectrum for any given distribution of droplet diameters at any cooling rate. IPA uses as input the distribution of droplet diameters and a parametrization of nucleation rate J(T) from the literature (Fig 13). IPA includes various previously reported parameterizations and can be extended to use others introduced by the users. We have demonstrated the application of IPA in predicting the impact of droplet diameter distributions typical of clouds on the evolution of the fraction of frozen droplets with temperature. By integrating the cooling rate and size dependence into the ice nucleation rates, the results and tools provided in this study could be used to improve and test approximations made in cloud models.

Minor comments:

Title: the title does not really highlight that this manuscript describes a model study, nor that homogeneous ice nucleation was investigated. Suggestion: "Modeling homogeneous ice nucleation from drop-freezing experiments: Impact of volume dispersion and cooling rates"

We thank the reviewer for suggesting the improved title. We adopted the suggestion.

Abstract: Could you describe why homogeneous ice nucleation is important for atmospheric science?

We rewrote the abstract (see above) to clarify the importance of homogeneous ice nucleation in atmospheric science.

Line 3-5: Can you state what the model predicts? Nucleation rates? Survival probability? Predicts Kinetics?

The AINTBAD model, implemented in Python, parameterizes a model for ice nucleation rates by analyzing droplet freezing data in a way that eliminates the effects of noise from numerical differentiated rate estimates. The IPA model, also implemented in Python, accounts for droplet size dispersion and specified cooling rates while predicting the fraction of frozen droplets.

We have added Figure. 13 to page 20 to clarify the inputs and outputs of our calculations.

[Figure]

We also added this text in the conclusions:

Here we directly analyze the fraction of frozen droplets vs temperature data to estimate rate parameters within a stochastic survival probability framework. We implement our approach in a Python code, AINTBAD (Fig. 13) that extracts the prefactor A and barrier B parameters according to a classical nucleation theory (CNT)-type rate expression. An advantage of our method is that it does not require large number of droplets to estimate accurate nucleation rates at each temperature. Although AINTBAD does not directly use nucleation rates in the optimization, it yields accurate estimates of the nucleation rate and avoids the noise associated with numerical differentiation.

And also this text in the conclusions:

We develop the Python code IPA to predict the nucleation spectrum for any given distribution of droplet diameters at any cooling rate. IPA uses as input the distribution of droplet diameters and a parametrization of nucleation rate J(T) from the literature (Fig. 13). IPA includes various previously reported parameterizations and can be extended to use others introduced by the users. We have demonstrated the application of IPA in predicting the impact of droplet diameter distributions typical of clouds on the evolution of the fraction of frozen droplets with temperature.

Check for typos and missing words within the manuscript: For example, abstract line 3: "... we develop a model that accounts for ..."

We combed the manuscript carefully for typos. Thank you.

Equation (3), Tm only gets explained later. Please define Tm when talking about equation 3.

We added a sentence to clarify the meaning of $T_m$ on pg. 4: "… where $T_m$ is the melting temperature."

Line 86: It might be helpful to explain the physical origin of the classical nucleation theory in two sentences to understand where equation 5 comes from. Especially, since the factors in this equation are referred to later on.

We added a brief discussion of nucleation theory to pg. 4:

The nucleation rate J is a product of the equilibrium concentration of clusters of a critical size and the non-equilibrium flux to post-critical sizes. Classical nucleation theory predicts prefactors and exponential terms with explicit temperature dependencies.

The exponent in classical nucleation theory is interpreted as a Gibbs free energy barrier, $\Delta G^*/k_B T$. It depends explicitly on both the absolute temperature and on the supercooling. To account for the temperature dependence of nucleation and the time-dependent temperature, we use $\delta_T = (T_m - T)/T_m$ as the dimensionless temperature and rewrite the expression for J as

$J = A \exp[-B/[(1-\delta_T)\delta_T^2]]$

Here $B = 16\pi\gamma 3 v_0^2 / 3\lambda_f^2 k_B T_m$ contains shape factors, physical constants, the latent heat, and interfacial free energy.

These quantities are nearly independent of temperature for the narrow temperature range where homogeneous nucleation is observed in the experiments (Kashchiev (2000); Sear (2007); Koop et al. (2000)). Thus the parameter B should be a nearly temperature independent parameter, while the barrier $\Delta G^*/kBT$ is strong function of temperature because of the $1/\Delta T^2$ factor.

The prefactor is related to the frequency at which water molecules at the ice-water interface attach to the critical nucleus and to the number of ice molecules that must attach to surmount the barrier. The prefactor is proportional to the self-diffusivity of water, and therefore it depends on temperature. However, over the small range of nucleation temperatures in this study (ca. 2K) we assume that the prefactor is temperature independent.

Line 112: Do you refer to the droplets diameter when talking about the size? A clear definition would be helpful including the calculated volume (in $\mu m^3$ or pL).

Size variable d refers to droplet diameter, and we added this note to pg. 10:

The volume is computed assuming spherical droplets i.e. $V_0 = \pi d^3/6$, thus variations in diameter directly translate to changes in volume.

Line 137: How many droplets did Atkinson et al., 2016 measure?

We added this sentence on pg. 7:

A total of 581 droplets were used in the diameter range of 3.8–18.8 $\mu$m., an average of 96 droplets for each diameter.

Figure 3a and line 141: Why did you decide to only show the global fit and not the individual fits with different droplet sizes? How would the individual fit compare with the global fit?

We added our explanation as a sentence in the caption of Fig.3:

The individual fits are not shown because the curves overlap closely with the data and globally fitted model predictions.

Line 165: Very impressive to see that experiment-to-experiment variations results in more scattering in the predictions for A and B. Would a "perfect" experiment result in a dataset which gives the same values for A and B? Does the difference in A and B for different droplet sizes (Table 1) only originate from experimental uncertainty and errors?

In theory, a "perfect" experiment, which eliminates all sources of experimental uncertainty and errors, would result in a dataset where the values for parameters A and B are consistent across different measurements. In reality, subtle temperature dependences within A and B, combined with the fact that small droplets nucleate at lower temperatures, will lead to slightly different A and B values for droplets of different sizes even in a perfect experiment.

Figure 4: How would the pooled dataset overlap with the superposition model? Can the authors pool the data from Atkinson et al., 2016 and compare it with the model predictions?

We added this explanation at the beginning of section 5, to clarify that the superposition model is that which the reviewer described as a pooled analysis:

We used the term superposition for a data analysis that retains stratification in freezing temperature, but otherwise pools droplets together regardless of their size.

Line 215: What does sufficiently narrow mean? Is there a quantitative measure of how narrow the distribution is?

We added this text to clarify on pg. 11:

If the size ranges are sufficiently narrow relative to the mean, then the A and B parameters resulting from a fit to the superposition become identical to those from monodisperse droplets of the mean size. Fig. 6 shows that the relative width of the volume distribution is sufficient to predict the superposition error. Specifically, for less than 1% error in B (B_apparent/B_actual > 0.99) we must have DV/V < 0.25.

Figure 5: One data point is off the line (x=y), why?

Please see the revised Fig. 5. We had a mistake in these calculations! The earlier manuscript had obtained $V_{min}$ and $V_{max}$ from $\pi(d\pm\Delta d)^3/6$, which is not a uniform volume distribution centered on $\pi d^3/6$. We have corrected the issue in this version. Thank you very much for alerting us to the issue.

In line 225 you state that in Figure 7a we see data from Riechers at al., 2013. However, in the legend of Figure 7a you wrote Shardt et al., 2022. Please clarify.

Thank you for spotting this error. We changed the reference to Shardt et al. (2022)

Figure 7: It is hard to see the difference between this work and the literature. Could you consider having a second panel showing a "zoom-in" to the dataset, e.g. x-axis spanning from 235K to 240K and y-axis spanning 10^3 to 10^9

Per the reviewer's suggestion, we added (what is now) Figure 8b and its caption on pg. 14:

Panel b) shows the nucleation rate in the region from 235 to 240 K only.

Is there a reason why the cooling rate dependency is explained in section 7? If not, you might want to consider presenting the fitting of the cooling rate dependency (Shardt et al. 2022) prior the literature comparison of nucleation rates.

Per the reviewer's suggestion, we moved the section "Comparing homogeneous nucleation rate parametrizations" after the section on "Effect of cooling rate on nucleation parameters".

Line 256: Clarify why the results contradict the nucleation theory. Experimental uncertainty and errors?

Thank you for alerting us to confusing aspects of this section.  In section 5, line 209 we added this sentence to clarify:

If all temperature dependence comes from the free energy barrier $B/[(1-\delta_T)\delta_T^2]$, then large droplets that nucleate at milder supercoolings will also nucleate with higher free energy barriers.

Section 9.2 has also been entirely rewritten in the revised manuscript (pg. 16-17).  Here is a summary of the changes:

a.  We also took care to explain how the data in Fig. 7 and Fig. 3 seem to contradict expectations from classical nucleation theory.
b.  Finally, we explain the unexpected result by analyzing the temperature dependence of the supercooled water viscosity/diffusion which leads to a rather strong temperature dependence in the prefactor.

Line 296: You state a cooling rate of 1K/min, but the legend in Figure 10b says 1K/ns. Please correct.

Thank you for noticing this error.

We updated the units in the legend of Figure 10 b) and c) from K/ns to K/min.

Line 307: Why would you anticipate smaller delta G for smaller droplets and why is the opposite the case? This expectation arises because smaller droplets have a higher surface-to-volume ratio, leading to an increased influence of surface tension, which is thought to reduce the energy required for nucleation.

The ratio of surface to volume is not relevant for homogeneous nucleation, which depends only on the volume.  Also the surface tension involved in the nucleation is for the ice-liquid interface of the nucleus, and not the liquid-vapor interface of the droplet.  We added the following note about the effects of volume in homogeneous nucleation, just below eq. 1:

Note that J itself is independent of droplet volume, and accordingly parameters that define J should also be independent of volume.

We also added the following notes to clarify. On. pg. 9, paragraph 1:

As seen from Fig. 3 and as predicted in Eq. (7) large droplets in a broad distribution will nucleate early (at milder supercoolings), while small droplets will survive to deeper supercoolings. If all temperature dependence comes from the free energy barrier

$B/[(1-\delta_T)\delta_T^2]$, then large droplets that nucleate at milder supercoolings will also nucleate with higher free energy barriers.

and on pg. 16, section 9.2:

... the estimated Gibbs free energy barriers from the data of Shardt et al. are smaller than those estimated from the data of Atkinson et al. If $\Delta G^*/k_B T = B/(T \Delta T^2)$ with constant B, then $\Delta G^*$ should be larger for the droplets of Shardt et al. which nucleate at higher temperatures.

Line 336: "Through the HUB-backward code from de Almeida Ribeiro et al., (2023) ..."

Thank you for pointing out the typo. It has been fixed.

Line 322: -0.4K to +0.4K?

Thank you. We added the missing units on pg. 17:

"... -0.4 to +0.4 K (or -0.2 to 0.2 K)."

Conclusion: Could the authors explain in 1-2 sentences how the model could be developed further to predict heterogeneous freezing? What could we learn form applying the model to heterogeneous ice nucleation data, assuming very uniform ice nucleating particles (characteristic nucleation temperature) are measured in a defined concentration (defined surface area and thus nucleation site density)?

We added the following note to the conclusions

We restrict our discussion in this article to homogeneous nucleation, but it might be possible to develop similar methods for analysis of heterogeneous nucleation data. A key challenge is that pure water droplets vary only in volume, while heterogeneous nucleation sites may vary in surface chemistry, pore geometry, and size (area) of the active region. These differences lead to sites with different barriers and also different prefactors. Except for special cases of highly regular surfaces, the estimated A and B parameters will then reflect a superposition of survival probabilities from many different types of sites. To illustrate this point, we analyze the data of fraction of ice vs temperature for ice nucleation by Kaolinite from Zhang and Maeda (2022) using the AINTBAD code. The estimated barrier at $T_{50}$ = 267.2 K is approximately $2k_B T$ (Supp. Fig. S1), a low value indicative of a superposition with nucleation sites of many different barriers. Further developments are needed to disentangle the contributions of different sites in heterogeneous nucleation of ice.

**Referee #2:**

I support publication. The authors present a framework for analysis of homogeneous freezing data that addresses many of the experimental issues encountered in the sorts of experiments that are typically used in homogeneous freezing.

I was surprised to see just how much change the difference of a few microns in the diameter of a droplet in a freezing experiment can make in deriving the kinetic prefactor and the activation barrier. The most surprising point to me was just how big the energy barrier is at nucleation. If the free energy barrier is 77kT, as referenced for one of the runs in a dataset presented in the paper, the exponential in Eqn 12 is 3.6e-34. But the prefactor is so large that the nucleation rate is easily observable. I was always taught that the exponential factor (the energy barrier) is really the only thing to worry about because it is an exponential. This analysis clearly shows that the prefactor has to be considered. The authors specify the variables that make up the free energy (equn 13) or B (eqn 5). A is never specified that I can see, and this makes getting physical insight from the analysis more difficult. I was really intrigued to see Fig. 3b, showing that A and B are clearly correlated. The discussion of that point could be improved in the manuscript, and being able to see the parameters that go into each one would help. A and B should be correlated because they both depend on temperature; seeing it so clearly in Fig. 3b puts a fine point on it.

> We added a discussion of the kinetic prefactor on pg. 5

> The prefactor is related to the frequency at which water molecules at the ice-water interface attach to the critical nucleus and to the number of ice molecules that must attach to surmount the barrier.  The prefactor is proportional to the self-diffusivity of water, and therefore it depends on temperature.  However, over the small range of nucleation temperatures in this study (ca. 2K) we assume that the prefactor is temperature independent.

I would also appreciate a more comprehensive discussion of the variation of the parameters with volume. I think that B is independent of volume, but that it seems to vary with volume because A does change with volume. Because of that, the nucleation rate becomes higher or lower, shifting the observed temperature range over which nucleation is observed, which means that the observed free energy barrier is different.

> Classical nucleation theory gives the nucleation on a per volume per time basis, with parameters A and B that are both independent of droplet volume. For the nucleation rate on a per droplet per time basis, we multiply the nucleation rate J by the droplet volume V.  See the discussion below eq (1).  To emphasize that the independence extends to A and B we added this text after eq (1):

> Note that J itself is independent of droplet volume, and accordingly parameters that define J should also be independent of volume.

Minor points

Line 73: "p" in ln(p(t|V) should be capitalized, I think

> Thank you.  Capitalized P is updated in page 4 of the revised manuscript

Line 89: "... latent heat of ice,..." ice doesn't have a latent heat. I would rephrase to latent heat of freezing or latent heat of fusion.

Thank you.  We corrected the terminology on page 4.

Line 306: "…one might anticipate smaller DELTA G for smaller droplets…" why might one expect this? What parameter in the expression for the free energy difference depends on the size of the droplet?

The effect of droplet volume on DG is not explicit prediction of the theory, but rather emerges as an indirect consequence of the experimental process.  Small droplets survive to more deeply supercooled temperatures than large droplets, because they offer less volume for nucleation at same conditions. Because, smaller droplets tend to nucleate at lower temperatures, they tend to have smaller nucleation barriers at the moment of nucleation.  We added the following notes to clarify.  On. pg. 9, paragraph 1:

As seen from Fig. 3 and as predicted in Eq. (7) large droplets in a broad distribution will nucleate early (at milder supercoolings), while small droplets will survive to deeper supercoolings.

On pg. 16, section 9.2:

… the estimated Gibbs free energy barriers from the data of Shardt et al. are smaller than those estimated from the data of Atkinson et al.  If $\Delta G^*/k_B T = B/(T\,\Delta T^2)$ with constant A and B, then $\Delta G^*$ should be larger for the droplets of Shardt et al. which nucleate at higher temperatures.

And we also added a note on this point at line 209:

If all temperature dependence comes from the free energy barrier $B/[(1-\delta_T)\delta_T^2]$, then large droplets that nucleate at milder supercoolings will also nucleate with higher free energy barriers.

Lines 317,318: I think there's a pair of parentheses missing in the reference.

Thank you.  See added parentheses around group of references near line 330 in the revision.  Also see typo correction for missing word "from" on line 317.

**Note to both reviewers:**

We found and fixed two transposed entries in the rightmost column of Table 2.